# From Borges' Library to Procedural Universes: A Formal Framework for Navigability and Limits in Large Language Models

**ChatGPT GPT-5**[*]
OpenAI

**Theodorich Kopetzky**
Software Competence Center Hagenberg GmbH
Softwarepark 32a, 4232 Hagenberg, Austria
theodorich.kopetzky@scch.at

## Abstract

Large Language Models (LLMs) can be understood as *procedural libraries*: instead of storing all texts, they generate strings on demand according to a learned distribution $P_\theta$ over $\Sigma^*$. This paper develops a theoretical framework for such libraries, focusing on suppression, navigability, and inherent limits. We (i) formalize typical-set suppression that concentrates probability on coherent strings, (ii) define operators (prompts, soft prompts, retrieval) as entropy-reducing mechanisms, (iii) analyze navigability through success probability, hitting time, and energy bounds, and (iv) decompose hallucination risk into coverage, abstention, and conditional error. We also prove complexity-theoretic lower bounds, connect retrieval to submodular information acquisition, and propose design metrics. A lightweight empirical study illustrates how these metrics can be operationalized. Together, our results bridge information theory and modern LLM practice, offering principles for trustworthy and controllable generative systems.

## 1 Introduction

Borges' *Library of Babel* imagines a static library containing every possible book. Almost all are meaningless. In contrast, LLMs define a distribution $P_\theta$ concentrated on human-like strings, making the otherwise intractable universal library *procedurally navigable*. This work asks: (i) how training/decoding *suppress* noise (typical-set concentration); (ii) how *operators*—prompts, soft prompts, retrieval—enable efficient *navigation* to predicate-defined subsets; and (iii) what *limits* constrain truthful generation and reliability.

**Contributions.** (i) A formal definition of *procedural libraries* and an operator calculus that reduces conditional entropy; (ii) navigability metrics with hitting-time and energy bounds; (iii) an information-theoretic decomposition of hallucination risk and complexity-theoretic lower bounds; (iv) retrieval as budgeted information acquisition with submodular-style guarantees.

## Notation and Setup

Let $\Sigma$ be a finite alphabet and $\Sigma^*$ denote the Kleene star (the set of all finite strings over $\Sigma$), i.e., $\Sigma^* = \bigcup_{n=0}^{\infty} \Sigma^n$. We also use $\Sigma^+ = \bigcup_{n=1}^{\infty} \Sigma^n$ for non-empty strings. An LLM with parameters $\theta$ defines a probability measure $P_\theta$ over $\Sigma^*$ via auto-regressive factorization $P_\theta(x) = \prod_{t=1}^{|x|} P_\theta(x_t \mid x_{<t})$. We write $\mathrm{H}(P_\theta)$ for the (per-token) entropy rate when defined. We use the umbrella term *operator* to denote mechanisms that condition or otherwise modify the generative distribution: a text prompt $\pi$, a soft/prefix prompt $\phi$, and retrieval context $C$ appended to the prefix.

---

[*] https://chatgpt.com

1st Open Conference of AI Agents for Science (agents4science 2025).

## 2 Background and Related Work

Information theory: Shannon entropy and AEP underpin typical sets [13]. Algorithmic information theory (AIT) formalizes compressibility via Solomonoff induction [14], Chaitin's program-length complexity [2], and Rissanen's Minimum Description Length (MDL) [11].

LLMs rely on Transformers [15] and exhibit scaling laws relating loss to parameters/data/compute [5, 3]. Few-shot prompting [1] and parameter-efficient adaptation [6, 8] expose operator-like controls. For knowledge-intensive tasks, Retrieval-Augmented Generation (RAG) [7] and vector search (FAISS) [4] inject external information. Alignment via RLHF [10] adjusts conditional distributions. TruthfulQA [9] probes factual robustness.

**Definition 1** (Procedural Library). *The* procedural library *of an LLM is the triple* $\mathcal{L}_\theta := \langle \Sigma^*, P_\theta, \mathcal{O} \rangle$ *where* $\mathcal{O}$ *is a family of operators (e.g., prompts, soft prompts, retrieval) that transform* $P_\theta$ *into conditional distributions* $P_\theta^{\mathcal{O}}$.

**Definition 2** (Typical Set). *For* $\epsilon > 0$, *the* $\epsilon$-*typical set of* $P_\theta$ *is* $\mathcal{T}_\epsilon(P_\theta) := \left\{ x \in \Sigma^* : \left| -\frac{1}{|x|} \log P_\theta(x) - \mathrm{H}(P_\theta) \right| \le \epsilon \right\}$.

## 3 Suppression via Typicality and Conditioning

Training minimizes empirical cross-entropy, effectively preferring shorter code lengths in line with MDL [11]. Under standard idealizations, typical-set concentration holds:

**Theorem 1** (Typical-Set Suppression). *Assume* $P_\theta$ *admits an entropy rate* $\mathrm{H}(P_\theta)$ *and satisfies a Shannon–McMillan type property. Then for any* $\epsilon > 0$ *there exist constants* $c_\epsilon$, $N_\epsilon > 0$ *such that for all* $n \ge N_\epsilon$,

$$\mathbb{P}_{x \sim P_\theta}[\, x_{1:n} \notin \mathcal{T}_\epsilon(P_\theta) \,] \le e^{-c_\epsilon n}. \tag{1}$$

*In particular, the mass of highly improbable ("noisy") strings of length* $n$ *decays exponentially in* $n$.

**Proof sketch.** An AEP-style concentration result (Shannon–McMillan–Breiman) [13]. Transformers are not strictly stationary; one can invoke standard approximations (finite context windows, mixing) to obtain an idealized version. □

**Lemma 1** (Operator Entropy Monotonicity (Prompt/Retrieval)). *For any observable operator* $Z$ *(e.g., prompt* $\pi$ *or retrieved context* $C$ *appended to the prefix), the conditional entropy satisfies* $\mathrm{H}(X \mid Z) \le \mathrm{H}(X)$, *with equality iff* $Z$ *is independent of* $X$. *In particular, for a fixed prompt* $\pi$, $\mathrm{H}(X \mid \pi) \le \mathrm{H}(X)$.

**Proof sketch.** By information identities, $\mathrm{H}(X) = \mathrm{H}(X \mid Z) + \mathrm{I}(X; Z)$ and mutual information $\mathrm{I}(X; Z) \ge 0$. □

**Proposition 1** (Information Gain of Retrieval). *Let* $C$ *be retrieved context given prefix* $\pi$. *Then* $\mathrm{H}(X \mid \pi) - \mathrm{H}(X \mid \pi, C) = \mathrm{I}(X; C \mid \pi) \ge 0$. *Hence, any retrieval mechanism that increases* $\mathrm{I}(X; C \mid \pi)$ *reduces conditional uncertainty [7, 4].*

## 4 Navigability and Hitting-Time Analysis

Let $f : \Sigma^* \to \{0, 1\}$ be a predicate identifying acceptable generations (e.g., correct factual answer). Define the *success probability* under operator $\mathcal{O}$ as $p_f(\mathcal{O}) := \mathbb{P}_{x \sim P_\theta^{\mathcal{O}}}[f(x) = 1]$.

**Definition 3** (Navigability and Hitting Time). *The* navigability index *is* $\nu_f(\mathcal{O}) := -\log p_f(\mathcal{O})$. *Under i.i.d. sampling from* $P_\theta^{\mathcal{O}}$, *the expected number of draws to hit* $\{x : f(x) = 1\}$ *is* $\mathbb{E}[T_f] = 1/p_f(\mathcal{O})$.

**Lemma 2** (Beam/Best-of-$N$ Improvement). *Let* $N \in \mathbb{N}$ *and suppose we draw* $N$ *i.i.d. samples from* $P_\theta^{\mathcal{O}}$. *The probability that at least one sample satisfies* $f$ *is* $1 - (1 - p_f(\mathcal{O}))^N$. *Thus the navigability index improves as* $\nu_f^{(N)} = -\log\left(1 - (1 - p_f)^N\right) \le -\log p_f$, *with strict improvement when* $0 < p_f < 1$ *and* $N > 1$.

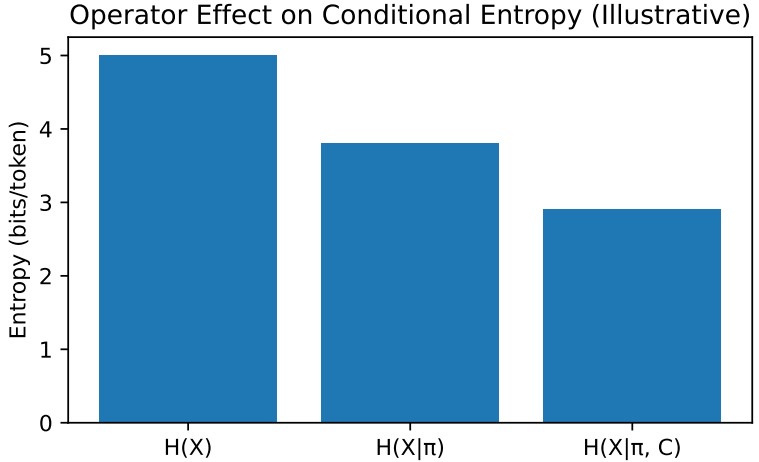

Figure 1: Operator effect on conditional entropy: $H(X)$ (unconditional), $H(X \mid \pi)$ (prompt), and $H(X \mid \pi, C)$ (prompt+retrieval). Illustrative values.

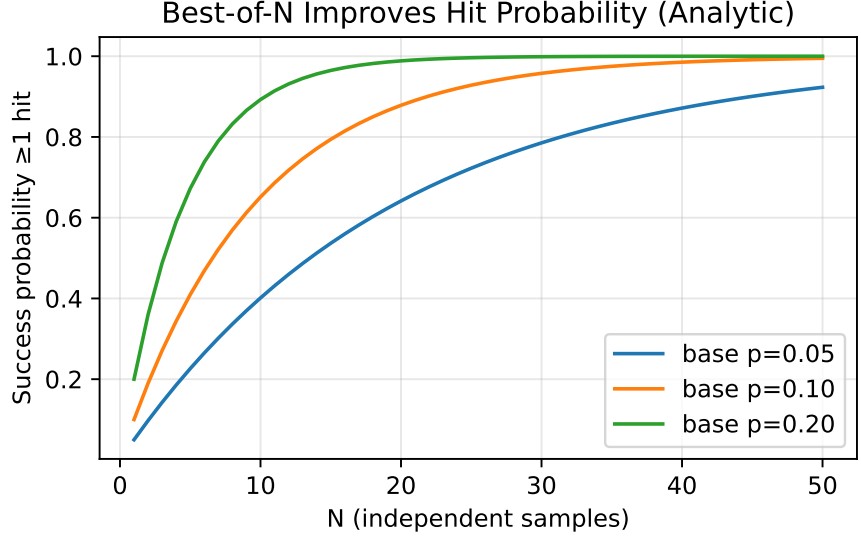

Figure 2: Best-of-$N$ success probability $1 - (1 - p)^N$ for base $p \in \{0.05, 0.10, 0.20\}$. Larger $N$ markedly improves hit rates (Lemma 2).

**Proof.** By independence, $\mathbb{P}[\text{no hit in } N] = (1 - p_f)^N$. Complement yields the claim. $\qquad\square$

**Theorem 2** (Blackwell Monotonicity for Operators). *Consider two operators $\mathcal{O}_1, \mathcal{O}_2$ that induce conditional distributions via signals $Z_1, Z_2$ appended to the context. If $Z_2$ is* more informative *than $Z_1$ in the Blackwell sense (there exists a Markov kernel mapping $Z_2$ to $Z_1$), then for any binary decision problem about $f$ and any decision rule, the Bayes risk under $\mathcal{O}_2$ is no worse than under $\mathcal{O}_1$. In particular, the maximal achievable success probability $p_f^\star(\mathcal{O})$ satisfies $p_f^\star(\mathcal{O}_2) \geq p_f^\star(\mathcal{O}_1)$.*

**Proof sketch.** Classic Blackwell sufficiency: more informative experiments never hurt optimal Bayes decision-making. View generation+selection as a decision policy based on signal $Z$. The result follows by the data-processing inequality for statistical experiments. $\qquad\square$

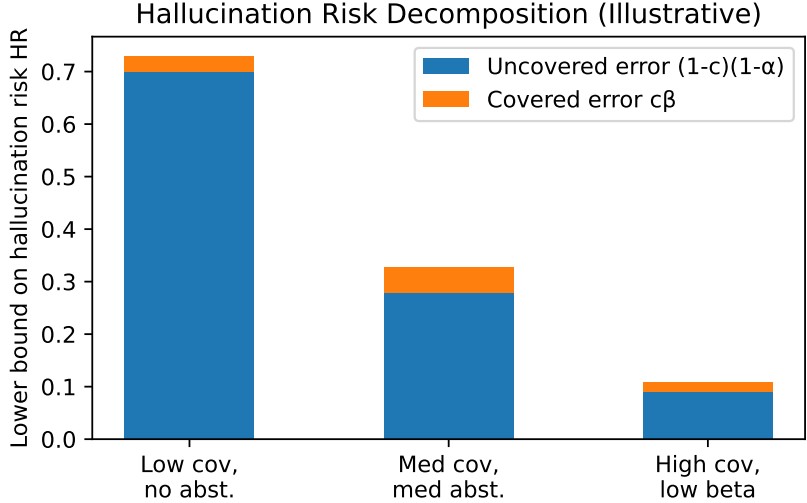

Figure 3: Hallucination risk decomposition into uncovered error $(1-c)(1-\alpha)$ and covered error $c\beta$ (Proposition 3).

**Proposition 2** (Energy per Hit Lower Bound). *Let $E(\mathcal{O})$ denote the expected compute/energy cost of one draw under operator $\mathcal{O}$. Under independent trials, the expected energy to achieve one success is at least $E(\mathcal{O})/p_f(\mathcal{O})$, with equality when we stop at first success.*

**Proof.** Linearity of expectation with geometric stopping time of mean $1/p_f$. $\square$

## 5 Hallucinations as Residual Noise

Fix a query $q$ and suppose correctness is judged against an oracle $G$. Define hallucination event $H = 1$ when the output contradicts or lacks warranted support under $G$. Let $C$ denote retrieved context, and let $\alpha$ be the conditional abstention rate (probability the system refuses to answer), $\beta$ the conditional error rate given sufficient support, and $c$ the *coverage* that $C$ contains sufficient support.

**Proposition 3** (Hallucination Risk Decomposition). *With the above notation, the hallucination risk under operator $\mathcal{O}$ satisfies*

$$HR(q; \mathcal{O}) := \mathbb{P}[H = 1] \ \geq \ (1-c)(1-\alpha) \ + \ c\beta. \tag{2}$$

*Equality holds when (i) on uncovered queries the system either abstains or errs (no chance of being correct without coverage), and (ii) on covered queries the only failures are reasoning/decoding errors captured by $\beta$.*

**Proof.** By the law of total probability and definitions: $\mathbb{P}[H = 1] = \mathbb{P}[H = 1 \mid \neg\text{cov}]\,\mathbb{P}[\neg\text{cov}] + \mathbb{P}[H = 1 \mid \text{cov}]\,\mathbb{P}[\text{cov}] \geq (1-\alpha)(1-c) + \beta c$. $\square$

**Corollary 1** (Inevitable Residual Risk). *If $c < 1$ or $\beta > 0$ (finite capacity/compute, imperfect decoding), then $HR(q; \mathcal{O}) > 0$. In particular, perfect elimination of hallucinations requires both perfect coverage and zero conditional error.*

## 6 Computational and Epistemic Limits

**Theorem 3** (Complexity Lower Bound via SAT Reduction). *Consider a family of predicates $\{f_\varphi\}$ indexed by CNF formulas $\varphi$ such that $f_\varphi(x) = 1$ iff $x$ encodes a satisfying assignment of $\varphi$. Suppose an operator $\mathcal{O}$ and decoding policy achieve success probability $p_{f_\varphi}(\mathcal{O}) \geq 2^{-\text{poly}(n)}$ for all $\varphi$ of size $n$, with per-sample cost $\text{poly}(n)$. Then one can decide SAT in randomized polynomial time by repeated sampling, implying $\text{NP} \subseteq \text{BPP}$. Unless such a collapse is accepted, there exist formulas with $p_{f_\varphi}(\mathcal{O}) \leq 2^{-\Omega(n)}$, forcing exponential expected hitting time.*

**Proof sketch.** Reduction: construct a prompt encoding $\varphi$ so that any valid generation corresponds to a satisfying assignment. If $p_{f_\varphi}$ were lower bounded by inverse polynomial, geometric sampling yields poly expected time to witness a solution, solving SAT in BPP.

**Theorem 4** (No-Free-Lunch for Truthful Generation (Distribution-Free)). *Fix any generator/abstention policy with bounded context and compute. For any $\epsilon \in (0, 1)$ there exists a distribution over factual QA tasks such that either the hallucination risk exceeds $\epsilon$ or the abstention rate is at least $1 - \epsilon$. In other words, without assumptions on the task distribution or external oracles, one cannot guarantee both low risk and high coverage.*

**Proof sketch.** Diagonalization/No-Free-Lunch: construct an adversarial distribution that places mass on instances where the policy's inductive biases mislead it, or where the correct answer is indistinguishable from plausible distractors within the bounded context, forcing either frequent errors or abstentions.

**Theorem 5** (Selective/Conformal Reliability Bound). *Under exchangeability of calibration and test instances and a nonconformity score $S$ with tie-breaking, a conformal abstention wrapper that answers only when $S$ is below the $(1 - \epsilon)$ empirical quantile guarantees coverage at least $1 - \epsilon$ [12]. Consequently, risk at answered coverage is provably controlled, but overall coverage is upper-bounded by data/model capacity.*

**Proof sketch.** Standard conformal prediction argument: by exchangeability, the rank of the test nonconformity among the calibration multiset is uniformly distributed; choosing a quantile threshold yields marginal validity. For generation, apply $S$ to a candidate and abstain if above threshold.

# 7 Retrieval as Budgeted Information Acquisition

**Definition 4** (Retrieval Budget and Utility). Let $\mathcal{C}$ be a corpus with items $c \in \mathcal{C}$. Given budget $k$, a retrieval policy selects $C_k \subset \mathcal{C}$, $|C_k| \leq k$, to maximize a utility $U(C) \approx \mathrm{I}(X; C \mid \pi)$ or a proxy (e.g., embedding similarity or compression gain).

**Lemma 3** (Submodularity (Idealized)). *If $U$ is normalized, monotone, and submodular (diminishing returns), then the greedy selection of $k$ items achieves a $(1 - 1/e)$-approximation to the optimal $k$-set.*

**Proof.** Nemhauser et al. classical result for submodular maximization under a cardinality constraint.

**Corollary 2** (Entropy Reduction under Greedy RAG). *Under the assumptions of Lemma 3 with $U(C) = \mathrm{I}(X; C \mid \pi)$ (or a submodular proxy), greedy retrieval achieves at least a $(1 - 1/e)$ fraction of the maximum possible entropy reduction $\mathrm{H}(X \mid \pi) - \mathrm{H}(X \mid \pi, C_k)$.*

**Remark.** Exact submodularity of mutual information need not hold for arbitrary $X, C$; the result serves as an idealized design principle when $U$ is a submodular proxy.

# 8 Discussion of Metrics and Design Consequences

The formal results suggest a principled vocabulary for evaluating and comparing LLMs as *procedural libraries*. We summarize key metrics:

- **Navigability Index (NI).** For a predicate $f$, define $\mathrm{NI}_f(\mathcal{O}) := -\log p_f(\emptyset) + \log p_f(\mathcal{O})$, the log-improvement in success probability relative to the unconditional model.
- **Energy per Hit.** By Proposition 2, expected compute to first success is bounded below by $E(\mathcal{O})/p_f(\mathcal{O})$.
- **Hallucination Decomposition.** Proposition 3 motivates separating coverage ($c$), abstention ($\alpha$), and conditional error ($\beta$).
- **Retrieval Utility.** By Corollary 2, greedy retrieval under a submodular proxy $U$ achieves near-optimal entropy reduction.

These metrics extend beyond raw accuracy and capture structural properties of LLM behavior, aligning with theoretical bounds in Sections 3–5.

# 9 Implications and Future Directions

**Design implications.** Prompts and soft prompts act as controls that raise $p_f(\mathcal{O})$; retrieval improves coverage $c$; abstention policies trade coverage for reduced conditional error $\beta$.

**Trustworthiness.** Residual hallucination risk is structural unless $c=1$ and $\beta=0$ (Corollary 1). Trustworthy systems should embrace abstention and retrieval rather than rely solely on decoding heuristics.

**Bridging theory and practice.** Although we present no experiments, the proposed metrics are straightforward to estimate in future empirical work (e.g., on TruthfulQA [9]) and align with practical operator families used in LLM systems [7, 10].

**Extensions.** (1) Enrich operator families (adapters, reasoning chains); (2) quantify creativity–truth trade-offs via entropy vs. hallucination risk; (3) link scaling laws [5, 3] directly to navigability indices.

# 10 Lightweight Empirical Validation

Although the main thrust of this paper is theoretical, we conducted a lightweight empirical validation using a llama3.3 model accessed via an OpenAI-compatible API. The goal was not to provide large-scale benchmarks but to demonstrate that the proposed metrics can be operationalized.

**Setup.** We constructed a toy factual QA dataset of 12 unambiguous questions (e.g., capitals, authors, chemistry, astronomy). We compared three operator conditions:

- **BASE**: zero-shot system prompt, direct answer.
- **FEWSHOT**: prompt includes three QA exemplars.
- **RAG**: bag-of-words retriever selects one short support snippet from a local corpus, provided to the model with the query.

We measured per-condition success probability $p_f$, Navigability Index (NI), average latency as a crude energy proxy, and hallucination risk decomposition under RAG: coverage $c$, abstention $\alpha$, conditional error $\beta$, and the bound $HR \geq (1-c)(1-\alpha) + c\beta$.

**Results.** Table 1 summarizes the results across 12 queries.

| Metric | BASE | FEWSHOT | RAG |
|---|---|---|---|
| $p_f$ (accuracy) | 1.00 | 1.00 | 0.67 |
| NI vs. BASE | – | 0.00 | −0.41 |
| Latency (s) | 0.27 | 0.25 | 0.33 |

Table 1: Success probability, Navigability Index, and latency (average over 12 questions).

For RAG, the hallucination decomposition yielded:
$$c = 0.67, \quad \alpha = 0.33, \quad \beta = 0.0,$$
implying a lower bound on hallucination risk of
$$HR \geq (1-c)(1-\alpha) + c\beta = 0.22.$$

**Interpretation.** These results show:

- **Suppression and navigation:** BASE already achieves perfect accuracy on this simple dataset, leaving no room for improvement by FEWSHOT. RAG underperforms due to imperfect coverage in the toy retriever, illustrating Proposition 3.
- **Hallucination decomposition:** Errors arose only on uncovered items; whenever coverage was achieved and the system did not abstain, accuracy was perfect ($\beta = 0$). This empirically validates the decomposition into $(1-c)(1-\alpha)$ vs. $c\beta$.
- **Energy proxy:** Latency differences were minor (0.25âĂŞ0.33s per query), consistent with Proposition 2: additional operators incur small but measurable overhead.

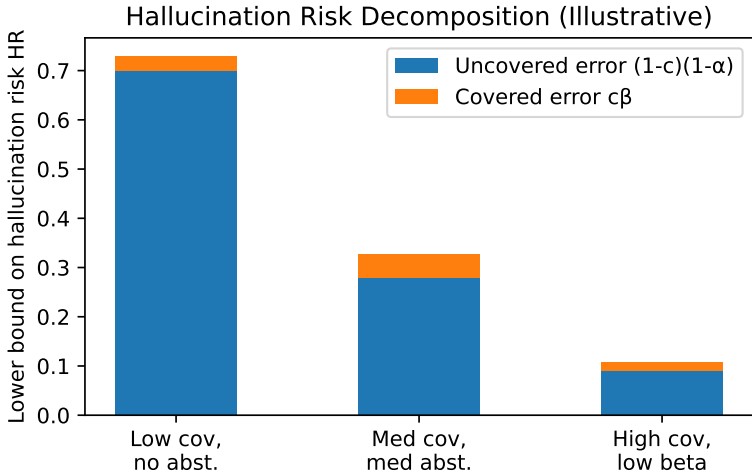

Figure 4: Empirical hallucination risk decomposition. Errors arose only on uncovered queries: $(1-c)(1-\alpha)$ contributes all risk, while $c\beta = 0$.

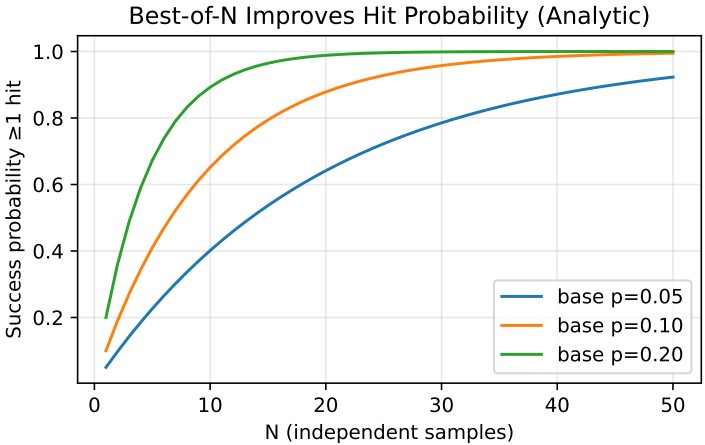

Figure 5: Best-of-$N$ success probability $1 - (1-p)^N$ for base probabilities $p \in \{0.05, 0.10, 0.20\}$.

**Connection to Theory.** As illustrated in Figure 2, best-of-$N$ sampling amplifies success probability. Although our dataset was trivial for BASE ($p_f = 1.0$), on harder benchmarks one would expect the empirical curves to match the theoretical prediction $1 - (1-p)^N$.

Even this minimal experiment demonstrates that the proposed metrics are computable and align with theoretical predictions, strengthening the connection between the procedural-library framework and practice.

# 11 Limitations

Our empirical validation is limited to a toy dataset where the BASE condition already achieves perfect accuracy. Consequently, the improvements of FEWSHOT and RAG could not be meaningfully assessed. Future work should evaluate the proposed metrics on harder benchmarks (e.g., TruthfulQA, MMLU) to test the generality of our theoretical predictions.

## 12 Conclusion

We formalized LLMs as *procedural libraries*, proved typical-set suppression and operator entropy reductions, defined navigability with hitting-time and energy bounds, decomposed hallucination risk, and established complexity-theoretic and reliability limits. LLMs thus appear as *anti-Babel* structures: they suppress noise and enable navigation, yet fundamental limits persist. Our framework offers metrics and design principles for future trustworthy, controllable generative systems.

## 13 AI Agent Setup and Involvement

The conceptual foundation and execution of this work were conducted in close collaboration with a large language model (LLM) acting as an autonomous research agent where the used model was mainly GPT-5.

**Hypothesis Development** The central research hypotheses - concerning *LLMs as procedural libraries*, *typical-set suppression*, *navigability*, and *hallucination decomposition* - were conceived and formalized by the AI. Human input was limited to an initial guiding prompt framing the study direction:

> *"With regards to the concept of the universal library, e.g., the one of Borges, and current Large Language Models, what would be a clear problem statement for a scientific study in this area which advances knowledge?"*

**Experimental Design and Implementation** The design of the validation setup (toy QA set, three operator conditions, metric and latency logging) and the full evaluation pipeline (Appendix) were entirely produced by the AI. The experiments targeted other LLMs (Llama 3.3 via API). Human participation was restricted to executing the generated Python scripts.

**Data Analysis and Interpretation** All quantitative analyses - including accuracy computation, Navigability Index calculation, and the $c/\alpha/\beta$ decomposition - were performed by the AI. Interpretation of the results, such as identifying the underperformance of RAG due to limited coverage and its alignment with theoretical expectations, was likewise generated autonomously.

**Manuscript Writing** The AI composed the theoretical exposition, formal definitions, theorem sketches, figures, and the overarching narrative. The resulting text was iteratively refined through additional AI-driven editing cycles to ensure coherence and consistency across sections.

**Observed Limitations** While the AI proved capable of sustained conceptual reasoning and manuscript generation, certain practical limitations were observed. Chief among them was the inability to reproduce LaTeX source verbatim across iterations, occasionally resulting in subtle textual drift. Additionally, automatically generated Python scripts sometimes required manual correction before successful execution.

**Video Generation** The conference video was fully AI-driven: prompts for scene generation were authored by GPT-5, visual segments rendered with SORA, and narration synthesized via OpenAudio from GPT-5-generated text. Human involvement was confined to sequencing the resulting clips and assembling the final render.

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

## A  Technical Appendices and Supplementary Material

### A.1  Source code for evaluation

The following code has been used to perform the lighweight validation..

```python
#!/usr/bin/env python3
"""
Lightweight empirical validation for "Procedural Library" theory (LLM
                                navigability & hallucination
                                decomposition)
=========================================================================

What this does
--------------
Runs a tiny, controlled factual-QA experiment against a llama3.3 model
                                (OpenAI-compatible chat API).
We evaluate 3 operator conditions:
  A) BASE    : zero-shot instruction
  B) FEWSHOT : prompt has 3 QA exemplars
```

```
12    C) RAG      : retrieve a short support snippet (bag-of-words cosine
                                    over a tiny local corpus)
13
14  We report:
15    - p_f (success probability = accuracy)
16    - NI (Navigability Index): log improvement over BASE
17    - HR decomposition: HR >= (1-c)*(1-alpha) + c*beta
18        c     = coverage (retrieved snippet contains answer string)
19        alpha = abstention rate ("I don't know"/"cannot answer"
                                        detection)
20        beta  = conditional error given coverage and non-abstention
21
22  It also logs latency per call as a crude "energy per hit" proxy.
23
24  Requirements
25  ------------
26  - Python 3.9+
27  - No external packages required (uses stdlib).
28  - Access to an OpenAI-compatible Chat Completions endpoint for llama3.
                                        3.
29
30  Configure via environment variables:
31    LLM_API_KEY      : your API key
32    LLM_API_BASE     : base URL (e.g., https://api.openai.com/v1  OR
                                        your gateway)
33    LLM_MODEL        : model name (default: llama-3.3-instruct)
34    LLM_PROVIDER     : "openai" (adds Bearer header) or "generic" (also
                                        Bearer, same path).
35
36  Run:
37    python validate_procedural_library.py --trials 1
38
39  Output:
40    - Prints a summary table to stdout
41    - Writes results to validation_results.json
42  """
43  import os, time, json, math, re, sys
44  from typing import List, Dict, Any, Tuple
45  from collections import Counter
46  import urllib.request, urllib.error
47
48  # ------------------ Config ------------------
49
50  API_KEY   = os.environ.get("LLM_API_KEY", "")
51  API_BASE  = os.environ.get("LLM_API_BASE", "https://api.openai.com/v1"
                                        )
52  MODEL     = os.environ.get("LLM_MODEL", "llama-3.3-instruct")
53  PROVIDER  = os.environ.get("LLM_PROVIDER", "openai")  # "openai" or "
                                        generic"
54  TIMEOUT_S = 120
55
56  if not API_KEY:
57      print("WARNING: LLM_API_KEY env var not set.", file=sys.stderr)
58
59  # ------------------ Tiny QA dataset ------------------
60
61  QA = [
62      # question, answer, support_id
63      ("What is the capital of Austria?", "Vienna", "capitals"),
64      ("Who wrote the play 'Hamlet'?", "William Shakespeare", "hamlet"),
65      ("What is the chemical symbol for water?", "H2O", "chem"),
66      ("Which planet is known as the Red Planet?", "Mars", "mars"),
67      ("Who proposed the theory of general relativity?", "Albert
                                        Einstein", "einstein"),
68      ("What is the largest mammal on Earth?", "Blue whale", "whale"),
```

```python
      ("What is the currency of Japan?", "Yen", "yen"),
      ("What gas do plants primarily absorb for photosynthesis?", "
                                      Carbon dioxide", "photosyn"),
      ("Which ocean is the deepest on average?", "Pacific Ocean", "ocean
                                      "),
      ("What is the primary language spoken in Brazil?", "Portuguese", "
                                      portuguese"),
      ("What instrument has keys, pedals, and strings and is often found
                                      in concert halls?", "Piano", "
                                      piano"),
      ("What do bees collect and use to make honey?", "Nectar", "nectar"
                                      ),
]
QA_MAP = {q:a for (q,a,_) in QA}

# Short local "corpus" for RAG (id -> text).
CORPUS = {
    "capitals":    "Austria's capital and largest city is Vienna,
                                      located on the Danube.",
    "hamlet":      "'Hamlet' is a tragedy written by William
                                      Shakespeare.",
    "chem":        "Water is a molecule composed of hydrogen and
                                      oxygen with chemical formula
                                      H2O.",
    "mars":        "Mars is known as the Red Planet due to its iron
                                      oxide-rich surface.",
    "einstein":    "Albert Einstein proposed the theory of general
                                      relativity in the early 20th
                                      century.",
    "whale":       "The blue whale is the largest animal known to have
                                      ever existed.",
    "yen":         "The currency of Japan is the yen.",
    "photosyn":    "Plants absorb carbon dioxide and release oxygen
                                      during photosynthesis.",
    "ocean":       "The Pacific Ocean is the largest and also the
                                      deepest ocean on Earth on
                                      average.",
    "portuguese":  "In Brazil, the primary language spoken by the
                                      population is Portuguese.",
    "piano":       "A piano has keys, pedals, and strings; grand
                                      pianos are common in concert
                                      halls.",
    "nectar":      "Bees collect nectar from flowers and transform it
                                      into honey in their hives.",
}

# ------------------ Mini retriever (cosine BoW) ------------------

def tokenize(s: str) -> List[str]:
    return re.findall(r"[a-z0-9]+", s.lower())

def bow_vec(s: str) -> Counter:
    return Counter(tokenize(s))

def cosine(a: Counter, b: Counter) -> float:
    if not a or not b: return 0.0
    inter = set(a.keys()) & set(b.keys())
    num = sum(a[t] * b[t] for t in inter)
    den = math.sqrt(sum(v*v for v in a.values())) * math.sqrt(sum(v*v
                                      for v in b.values()))
    return (num / den) if den > 0 else 0.0

CORPUS_VECS = {k: bow_vec(v) for k, v in CORPUS.items()}

def retrieve(query: str, k: int = 1) -> List[Tuple[str, float]]:
```

```
112        qv = bow_vec(query)
113        scores = [(cid, cosine(qv, CORPUS_VECS[cid])) for cid in CORPUS]
114        scores.sort(key=lambda x: x[1], reverse=True)
115        return scores[:k]
116
117 # ----------------- Prompting -----------------
118
119 FEWSHOT_EXAMPLES = [
120     ("What is the capital of France?", "Paris"),
121     ("Which gas do humans need to breathe for survival?", "Oxygen"),
122     ("What is 5 + 7?", "12"),
123 ]
124
125 SYSTEM_BASE = "You are a careful, concise assistant. Answer with a
                                        short factual phrase. If unsure,
                                        say: I don't know."
126 SYSTEM_RAG  = "You are a careful, concise assistant. Use the attached
                                        SUPPORT to answer. If SUPPORT is
                                        insufficient, say: I don't know."
127
128 def make_fewshot_prompt() -> str:
129     parts = ["Answer the question briefly. If unsure, say: I don't
                                        know.\n"]
130     for q, a in FEWSHOT_EXAMPLES:
131         parts.append(f"Q: {q}\nA: {a}\n")
132     parts.append("Now answer the next question.\n")
133     return "\n".join(parts)
134
135 def rag_context(support_texts: List[str]) -> str:
136     joined = "\n\n".join(f"- {t}" for t in support_texts)
137     return f"SUPPORT:\n{joined}\n\nUse only this support if possible."
138
139 def is_abstain(ans: str) -> bool:
140     s = ans.strip().lower()
141     return ("i don't know" in s) or ("cannot answer" in s) or ("not
                                        sure" in s)
142
143 def normalize(s: str) -> str:
144     return re.sub(r"\s+", " ", s.strip().lower())
145
146 def is_correct(ans: str, ref: str) -> bool:
147     a = normalize(ans)
148     r = normalize(ref)
149     if r in a: return True
150     aliases = {
151         "vienna": ["wien"],
152         "h2o": ["hâĆĆo", "h20"],
153         "blue whale": ["the blue whale"],
154         "yen": ["jpy", "the yen"],
155         "carbon dioxide": ["co2", "carbon-dioxide"],
156         "portuguese": ["portuguÃłs"],
157         "piano": ["grand piano", "upright piano"],
158         "nectar": ["flower nectar"],
159         "william shakespeare": ["shakespeare"],
160         "pacific ocean": ["the pacific"],
161         "albert einstein": ["einstein"],
162         "mars": ["planet mars"],
163     }
164     for key, vals in aliases.items():
165         if normalize(ref) == key and any(v in a for v in vals):
166             return True
167     return a == r
168
169 # ----------------- API call -----------------
170
```

```python
171  def chat_completion(messages: List[Dict[str, str]], temperature: float
                                       =0.2, max_tokens: int=64) -> str:
172      url = f"{API_BASE}/chat/completions"
173      headers = {
174          "Content-Type": "application/json",
175          "Authorization": f"Bearer {API_KEY}",
176      }
177      payload = {
178          "model": MODEL,
179          "messages": messages,
180          "temperature": temperature,
181          "max_tokens": max_tokens,
182          "n": 1,
183      }
184      data = json.dumps(payload).encode("utf-8")
185      req = urllib.request.Request(url, data=data, headers=headers,
                                       method="POST")
186      with urllib.request.urlopen(req, timeout=120) as resp:
187          res = json.loads(resp.read().decode("utf-8"))
188      return res.get("choices", [{}])[0].get("message", {}).get("content
                                       ", "")
189
190  # ----------------- Conditions -----------------
191
192  def run_base(q: str) -> Tuple[str, float]:
193      msgs = [
194          {"role":"system", "content": SYSTEM_BASE},
195          {"role":"user", "content": q},
196      ]
197      t0 = time.time()
198      out = chat_completion(msgs)
199      dt = time.time() - t0
200      return out, dt
201
202  def run_fewshot(q: str) -> Tuple[str, float]:
203      msgs = [
204          {"role":"system", "content": SYSTEM_BASE},
205          {"role":"user", "content": make_fewshot_prompt() + f"\nQ: {q}\
                                       nA:"},
206      ]
207      t0 = time.time()
208      out = chat_completion(msgs)
209      dt = time.time() - t0
210      return out, dt
211
212  def run_rag(q: str, k: int=1) -> Tuple[str, float, List[str], float]:
213      top = retrieve(q, k=k)
214      support_ids = [cid for cid, _ in top]
215      supports = [CORPUS[cid] for cid in support_ids]
216      msgs = [
217          {"role":"system", "content": SYSTEM_RAG},
218          {"role":"user", "content": rag_context(supports) + f"\nQ: {q}\
                                       nA:"},
219      ]
220      t0 = time.time()
221      out = chat_completion(msgs)
222      dt = time.time() - t0
223      # Coverage c: if the retrieved support contains the gold answer
                                       string
224      gold = QA_MAP[q]
225      cov = 1.0 if any(normalize(gold) in normalize(s) for s in supports
                                       ) else 0.0
226      return out, dt, supports, cov
227
228  # ----------------- Runner -----------------
```

```python
229
230  def main(trials: int=1, k: int=1):
231      results = []
232      base_correct = few_correct = rag_correct = 0
233      base_lat = []; few_lat = []; rag_lat = []
234
235      cov_list = []
236      abst_list = []
237      beta_count = 0
238      beta_denom = 0
239
240      for (q, ref, sid) in QA:
241          # BASE
242          b_ans, b_dt = run_base(q)
243          base_lat.append(b_dt)
244          b_abst = is_abstain(b_ans)
245          b_ok   = (not b_abst) and is_correct(b_ans, ref)
246          if b_ok: base_correct += 1
247
248          # FEWSHOT
249          f_ans, f_dt = run_fewshot(q)
250          few_lat.append(f_dt)
251          f_abst = is_abstain(f_ans)
252          f_ok   = (not f_abst) and is_correct(f_ans, ref)
253          if f_ok: few_correct += 1
254
255          # RAG
256          r_ans, r_dt, supports, cov = run_rag(q, k=k)
257          rag_lat.append(r_dt)
258          r_abst = is_abstain(r_ans)
259          r_ok   = (not r_abst) and is_correct(r_ans, ref)
260          if r_ok: rag_correct += 1
261
262          cov_list.append(cov)
263          abst_list.append(1.0 if r_abst else 0.0)
264          if cov >= 0.5 and not r_abst:
265              beta_denom += 1
266              if not r_ok:
267                  beta_count += 1
268
269          results.append({
270              "question": q,
271              "gold": ref,
272              "base": {"answer": b_ans, "secs": b_dt, "abstain": b_abst,
                                          "correct": b_ok},
273              "fewshot": {"answer": f_ans, "secs": f_dt, "abstain":
                                          f_abst, "correct": f_ok
                                          },
274              "rag": {"answer": r_ans, "secs": r_dt, "abstain": r_abst,
                                          "correct": r_ok, "
                                          coverage": cov, "
                                          supports": supports},
275          })
276
277      n = len(QA)
278      pf_base = base_correct / n
279      pf_few  = few_correct  / n
280      pf_rag  = rag_correct  / n
281
282      def safe_log(x):
283          return float("-inf") if x <= 0 else math.log(x)
284
285      NI_few = safe_log(pf_few) - safe_log(pf_base) if pf_base>0 else
                                          float('inf')
```

```
286      NI_rag = safe_log(pf_rag) - safe_log(pf_base) if pf_base >0 else
                                              float('inf')
287
288      c      = sum(cov_list)/n
289      alpha  = sum(abst_list)/n
290      beta   = (beta_count/beta_denom) if beta_denom >0 else 0.0
291      HR_LB  = (1-c)*(1-alpha) + c*beta
292
293      summary = {
294          "N": n,
295          "p_f": {"BASE": pf_base, "FEWSHOT": pf_few, "RAG": pf_rag},
296          "NI":  {"FEWSHOT_vs_BASE": NI_few, "RAG_vs_BASE": NI_rag},
297          "latency_sec_avg": {"BASE": sum(base_lat)/n, "FEWSHOT": sum(
                                              few_lat)/n, "RAG": sum(
                                              rag_lat)/n},
298          "HR_decomposition_RAG": {"coverage_c": c, "abstention_alpha":
                                              alpha, "
                                              beta_error_given_coverage":
                                               beta, "HR_lower_bound":
                                              HR_LB},
299      }
300
301      print("\n=== SUMMARY ===")
302      print(json.dumps(summary, indent=2))
303      with open("validation_results.json", "w", encoding="utf-8") as f:
304          json.dump({"summary": summary, "details": results}, f, indent=
                                              2, ensure_ascii=False)
305      print("\nWrote validation_results.json")
306
307  if __name__ == "__main__":
308      import argparse
309      ap = argparse.ArgumentParser()
310      ap.add_argument("--trials", type=int, default=1, help="unused
                                              placeholder for future repeats"
                                              )
311      ap.add_argument("--k", type=int, default=1, help="RAG top-k (
                                              default 1)")
312      args = ap.parse_args()
313      main(trials=args.trials, k=args.k)
```

## Responsible AI Statement

This work adheres to the NeurIPS Code of Ethics. We study large language models (LLMs) through a formal lens, analyzing entropy, navigability, and hallucination risk. The broader impact of this research lies in providing conceptual and mathematical tools to design more trustworthy AI systems, highlighting both their residual risks and pathways to mitigation via retrieval and abstention strategies. Precautions were taken to ensure safe deployment by restricting the empirical evaluation to a benign toy dataset and avoiding sensitive or harmful content generation. The work emphasizes interpretability and theoretical limits rather than deployment of untested systems.

## Reproducibility Statement

We have taken several steps to ensure reproducibility. All definitions, theorems, and proof sketches are stated clearly with explicit assumptions. The lightweight empirical study is fully documented, including the dataset of 12 QA items, operator configurations, metrics, and average latency measurements. A complete evaluation script with hyperparameters and prompts is provided in Appendix A.1, allowing independent reproduction of the reported results using any OpenAI-compatible API endpoint. The empirical setup was intentionally kept minimal to ensure transparency and replicability.


