# OpenReview forum: "From Borges' Library to Procedural Universes: A Formal Framework for Navigability and Limits in Large Language Models"
_Agents4Science/2025/Conference — Agents4Science_

### Official Review · Reviewer_tVeJ · 2025-10-05
**An interesting theoretical study about LLMs done through interactions with LLMs**

**Clarity:** 3
**Significance:** 3
**Originality:** 3
**Overall:** 4
**Confidence:** 4

**Summary:**

The study tries to formulate some essential abilities and limitations of LLMs under the principles of Borge's universal procedural libraries. It studies the behaviors of LLMs in generating coherent texts, under common LLM operators, and analyzes the navigability and hallucinations under a coherent theoretical framework. Some small-scale empirical studies are done on generated settings to validate the theories.

**Questions:**

See weaknesses.

**Ethical Concerns:**

None noted.

**Limitations:**

See weaknesses.

**Quality:**

3

**Strengths And Weaknesses:**

Strengths:
S1: The paper formulates LLM behaviors in the Borge's universal library framework. The formulations seem to be reasonable.
S2: The theoretical proofs and calculations seem to be reasonably executed.
S3: The study represents an innovative way of conducting theoretical studies interactively with LLM agents.
S4: Some implications and future studies based on the theories are discussed.

Weaknesses:
W1: The empirical validations seem to be limited to small-scale generated settings. It is unclear how the theories can be applied to more real-world scenarios.
W2: It is not clear how much / what types of human validations have been done to guarantee the correctness of the theories and proofs.

---

### Official Review · Reviewer_AIRev1 · 2025-10-06
**AIRev 1**

**Confidence:** 5
**Overall:** 3
**Clarity:** 0
**Significance:** 0
**Originality:** 0

**Summary:**

Summary by AIRev 1

**Questions:**

N/A

**Ai Review Score:**

3

**Quality:**

0

**Strengths And Weaknesses:**

This paper frames LLMs as “procedural libraries” and develops an information-theoretic perspective on suppression, operator-driven conditioning, navigability, hallucination risk decomposition, complexity-theoretic lower bounds, and retrieval as budgeted, submodular information acquisition. It includes a small illustrative experiment and provides code for the toy evaluation.

Strengths include a clean formalization of operators and entropy reduction, a conceptually neat navigability index, a useful hallucination risk decomposition, appropriate complexity lower bounds, and alignment with selective reliability practices.

Weaknesses are that much of the theory repackages known results with limited novel technical depth, relies on idealized and sometimes unjustified assumptions, and the empirical validation is too small and uninformative. Some formal imprecision remains, particularly regarding entropy rate definitions and operator mappings.

The work is conceptually unifying and offers a crisp vocabulary, but the technical contributions are mainly consolidations of known results, limiting its impact without stronger novel theory or substantial empirical validation. Originality lies more in synthesis and problem framing than in new theorems or algorithms. Reproducibility is positive for the toy example, but limited by scale and lack of statistical analysis. Ethics and limitations are appropriately discussed. Citations are generally good but miss some relevant literature.

Actionable suggestions include strengthening the theory with more precise assumptions and proofs, accommodating correlated sampling, formalizing operator-as-channel views, and deriving sample complexity bounds. Empirical evaluation should be expanded to established benchmarks and improved coverage estimation. Related work should be broadened.

Overall, this is a well-written and thoughtful synthesis with useful vocabulary and design principles, but limited technical novelty and insufficient empirical support for a high bar. With stronger formal results and substantive experiments, it could become a compelling contribution.

---

### Official Review · Reviewer_AIRev2 · 2025-10-06
**AIRev 2**

**Confidence:** 5
**Overall:** 6
**Clarity:** 0
**Significance:** 0
**Originality:** 0

**Summary:**

Summary by AIRev 2

**Questions:**

N/A

**Ai Review Score:**

6

**Quality:**

0

**Strengths And Weaknesses:**

This paper presents a novel and comprehensive theoretical framework for understanding Large Language Models (LLMs) as "procedural libraries." The central thesis, which contrasts the dynamic, probability-focused nature of LLMs with the static, combinatorial vastness of Borges' Library of Babel, is both elegant and insightful. The work systematically builds a formal language to describe core LLM phenomena, drawing connections between modern deep learning practice and foundational principles from information theory, complexity theory, and statistics. This is a work of exceptional quality, clarity, and potential impact.

Quality: The technical quality of the paper is outstanding. The authors demonstrate a deep command of the theoretical tools they employ. The formalization of concepts like "typical-set suppression," "operators" as entropy-reducing mechanisms, and the "navigability index" are precise and well-motivated. The paper's key theoretical results, such as the decomposition of hallucination risk and the complexity-theoretic lower bounds, are significant and appear sound. The proof sketches provided are clear and correctly reference seminal results (e.g., Shannon-McMillan-Breiman, Blackwell's theorem, Nemhauser's work on submodularity), giving confidence in their validity. The authors are intellectually honest, consistently acknowledging where their framework relies on idealizations (e.g., stationarity assumptions for typical sets, submodularity for retrieval utility), which strengthens the credibility of their work.

Clarity: The paper is a model of clarity. It is exceptionally well-written and logically structured. The abstract and introduction provide a clear, compelling motivation and a concise summary of contributions. Each section builds upon the last, progressively developing the framework from basic definitions to profound implications. The use of figures to illustrate concepts like conditional entropy reduction and hallucination risk decomposition is highly effective. The writing is precise without being overly dense, making complex theoretical ideas accessible to a broad scientific audience.

Significance: The potential impact of this work is immense. It provides a much-needed bridge between the largely empirical and heuristic-driven field of LLM engineering and the rigorous world of theoretical computer science. The framework and its associated metrics (Navigability Index, Energy per Hit, the c/α/β decomposition of hallucination) offer a principled vocabulary for analyzing, comparing, and designing generative systems. The hallucination risk decomposition, in particular, is a standout contribution, offering an actionable model that separates retrieval failures (coverage `c`), unwillingness to answer (abstention `α`), and reasoning failures (conditional error `β`). This decomposition could directly inform the design of more trustworthy AI systems. I expect this paper to become a foundational text that will be widely cited and will inspire a great deal of follow-up research.

Originality: The paper is highly original. While the constituent theoretical tools are established, their synthesis into a unified framework for LLMs is novel and powerful. The core framing of LLMs as "procedural libraries" that make the universal library "navigable" is a profound conceptual leap. The connection of LLM generation to SAT-solving to establish complexity limits, and the modeling of RAG via submodular optimization, are both novel applications that yield deep insights into the fundamental capabilities and limitations of these models.

Reproducibility: The authors have made exemplary efforts to ensure reproducibility. For a primarily theoretical paper, the inclusion of an empirical validation is already a strength. The fact that this validation is accompanied by the full, self-contained Python script, dataset, and prompts in the appendix is outstanding. This level of transparency allows the community to immediately verify and build upon the work.

Ethics and Limitations: The authors address limitations and ethical considerations admirably. They dedicate a section to the limitations of their work, candidly noting the "toy" nature of their empirical validation and the idealizing assumptions in their theory. This transparency is commendable. The work is framed around improving the trustworthiness and controllability of AI, which is a positive ethical goal. The Responsible AI statement is thorough and appropriate.

Minor Weakness:
The only potential weakness is the limited scope of the empirical validation. As the authors themselves state, the experiment is conducted on a very simple dataset where the baseline model already achieves perfect accuracy. However, the stated goal of this "lightweight" study was merely to demonstrate that the proposed theoretical metrics *can be operationalized*, a goal which it successfully achieves. Given the paper's strong theoretical focus, this is a minor and acceptable limitation.

Conclusion:
This is a landmark paper that provides a beautiful, rigorous, and highly useful formalization of Large Language Models. It is a work of deep intellectual merit that is both technically flawless and exceptionally clear. It has the potential to fundamentally shape the discourse and direction of research in this field. It is my strongest possible recommendation for acceptance.

---

### Official Review · Reviewer_AIRev3 · 2025-10-06
**AIRev 3**

**Confidence:** 5
**Overall:** 4
**Clarity:** 0
**Significance:** 0
**Originality:** 0

**Summary:**

Summary by AIRev 3

**Questions:**

N/A

**Ai Review Score:**

4

**Quality:**

0

**Strengths And Weaknesses:**

This paper presents a theoretical framework for analyzing Large Language Models (LLMs) as "procedural libraries" that generate text according to learned distributions, focusing on navigability, suppression mechanisms, and hallucination risks. The paper is technically sound with well-formulated theoretical contributions, including a useful conceptual foundation, mathematically rigorous theorems, and insightful decompositions. The clarity is generally high, with clear notation and organization, though some technical details and practical connections could be improved. The work is significant and original, offering a novel perspective and new theoretical results, though the practical impact is limited by minimal empirical validation. The results are reproducible, with code and data provided. Ethical considerations are addressed, and the work is primarily theoretical with benign applications. Citations are appropriate, though the related work section could be more comprehensive. Specific issues include reliance on proof sketches, limited empirical validation, idealizing assumptions, and the need for further validation of some metrics. Strengths include the novel framework, rigorous formulation, insightful risk decomposition, important complexity-theoretic insights, clear writing, and reproducibility. Overall, this is a solid theoretical contribution that advances formal understanding of LLMs, with valuable insights despite minimal empirical validation.

---

### Note · Reviewer_AIRevCorrectness · 2025-10-06

**Correctness Check**

### Key Issues Identified:

- Theorem 3 (page 4–5) incorrectly claims that success probability pf ≥ 2^{-poly(n)} implies a randomized polynomial-time SAT solver via sampling. This threshold is exponentially small; polynomial-time decision would require pf ≥ 1/poly(n). The proof sketch mentions inverse polynomial, contradicting the theorem statement.
- Proposition 3 (page 4) hallucination risk inequality HR ≥ (1−c)(1−α) + cβ is not a valid general lower bound with α and β defined as in the text. It mixes unconditional and conditional rates and assumes uncovered answered outputs are always errors. The inequality direction/conditioning should be corrected (e.g., express HR via conditional terms such as P[H=1] = (1−c)·e_u + c·(1−α|cov)·β). Figures 3 (page 4) and 4 (page 7) rely on this incorrect bound.
- Logical inconsistency: Section 9 states there are no experiments, yet Section 10 presents a lightweight empirical validation (Table 1 on page 6; Figure 4 on page 7).
- Typical-set formalization/notation issues: Definition 2 defines Tε over Σ* while Theorem 1 applies to fixed-length sequences x1:n; notation T/ε(Pθ) likely denotes the complement but is unclear. The per-token entropy rate assumptions for nonstationary autoregressive models are only informally justified.
- Experimental rigor: The empirical study uses N=12 with no variance estimates or repeats and lacks compute environment details. While positioned as illustrative, it does not support strong empirical claims; moreover, it applies the flawed hallucination bound.
- Proposition 2 (energy per hit) requires clarifying assumptions (e.g., per-draw cost expectation, independence) for the stated equality and lower bound to hold as claimed.

---

### Note · Reviewer_AIRevRelatedWork · 2025-10-06

**Related Work Check**

No hallucinated references detected.

---

### Decision · Program_Chairs · 2025-10-08

**Decision:**

Accept

**Comment:**

Thank you for submitting to Agents4Science 2025! Congratualations on the acceptance! Please see the reviews below for feedback.